# Multimorbidity, health service use, and health insurance by socioeconomic groups in 31 countries: A multi-cohort study

Yanshang Wang[1,2,3,4], Chang Cai[3], Zhenyu Shi[1,2,5], Qian Gao[3,6], Alex Bottle[3], Mansour Taghavi Azar Sharabiani[3], Joshua Stott[7], Benedict Hayhoe[3]*, Ping He[2,8,9,10]*

**1** School of Public Health, Peking University, Beijing, China, **2** China Center for Health Development Studies, Peking University, Beijing, China, **3** School of Public Health, Imperial College London, London United Kingdom, **4** Division of Psychiatry, Faculty of Brain Science, University College London, London, United Kingdom, **5** School of Social Development, University of Health and Rehabilitation Sciences, Qingdao National High- Tech Industrial Development Zone, Qingdao, Shandong, China, **6** Faculty of Population Health Sciences, University College London, London, United Kingdom, **7** Adaptlab, Department of Clinical and Educational Psychology, University College London, London, United Kingdom, **8** Beijing Institute of Health Development, Peking University, Beijing, China, **9** National Health Commission Key Laboratory of Health System Reform and Governance, Peking University, Beijing, China, **10** State Key Laboratory of Vascular Homeostasis and Remodeling, Peking University, Beijing, China

* b.hayhoe@imperial.ac.uk (BH); phe@pku.edu.cn (PH)

## Abstract

### Background

The prevalence of physical, psychological, and cognitive multimorbidity is characterised by marked socioeconomic status (SES) inequalities. However, the relationships between multimorbidity patterns—particularly those involving cognitive conditions—and healthcare utilisation, as well as the role of health insurance, remain poorly understood. This study aims to explore healthcare-seeking behaviour among individuals with multimorbidity and assess whether these associations vary by SES and health insurance coverage.

### Methods and findings

This multicohort study analysed harmonised data from six longitudinal studies across 31 countries, including participants aged 50 years and older. Multimorbidity was defined as the coexistence of two or more disorders across physical, psychological, or cognitive disorders. Outpatient and inpatient healthcare utilisation were measured. Random-effects logistic regression models were used to estimate associations with healthcare utilisation, and random-effects negative binomial models were applied to analyse visit frequencies. All models were adjusted for age, gender, educational attainment, work status, marital status, and SES, as well as lifestyle factors. Country-specific estimates were pooled using multinational meta-analysis to generate overall effect sizes.

**Data availability statement:** The data used in this study are publicly available from the Gateway to Global Aging Data (https://g2aging.org), which provides harmonised data from multiple nationally representative ageing surveys, including the Health and Retirement Study (HRS), Survey of Health, Ageing and Retirement in Europe (SHARE), China Health and Retirement Longitudinal Study (CHARLS), and others. Access to these data requires registration and approval from the respective data custodians. The analysis code supporting the findings of this study is available at: https://github.com/YanshangWang98/Multimorbidity-Healthcare-Service-Use.

**Funding:** This work was supported by Noncommunicable Chronic Diseases-National Science and Technology Major Project（Grant No.2023ZD0509601）and the Major Project of the National Social Science Fund of China (Grant No. 21&ZD187 to PH). The funders had no role in study design, data collection and analysis, decision to publish, or preparation of the manuscript.

**Competing interests:** The authors have declared that no competing interests exist.

**Abbreviations:** CHARLS, China Health and Retirement Longitudinal Study; CI, confidence interval; HRS, Health and Retirement Longitudinal Study; IRR, incident rate ratio; JSTAR, Japanese Study of Aging and Retirement; KLoSA, Korean Longitudinal Study of Aging; MHAS, Mexican Health and Aging Study; OR, odds ratio; SES, socioeconomic status; SHARE, Survey of Health, Ageing and Retirement in Europe; UHC, Universal Health Coverage.

Compared with individuals without any conditions, those with the most complex multimorbidity pattern (physical-psychological-cognitive multimorbidity) were more likely to use outpatient care (OR 3.21, 95% CI [2.39, 4.03]; $p < 0.001$) but not as high as those with physical-psychological multimorbidity (OR 7.84, 95% CI [6.59, 9.10]; $p < 0.001$). Additionally, the association varied across socioeconomic groups, individuals of lower SES experiencing more pronounced disparities in healthcare use. For inpatient care, adding a cognitive disorder to an existing physical or psychological condition was not associated with increased inpatient utilisation. Among individuals with health insurance coverage, the association between multimorbidity and outpatient care utilisation was generally attenuated. This was especially evident for those with physical-psychological-cognitive multimorbidity: insured individuals had an OR of 6.22 (95% CI [5.33, 7.25]; $p < 0.001$), compared with 3.40 (95% CI [3.03, 3.82]; $p < 0.001$) among uninsured individuals. A limitation of this study is that healthcare utilisation measures differed across cohorts and were harmonised retrospectively.

## Conclusions

Cognitive disorders further complicate the relationship between multimorbidity and health service use, indicating potential unmet healthcare needs, especially among individuals with lower SES. Our study highlights the potential role of health insurance in reducing socioeconomic disparities in healthcare utilisation associated with multimorbidity.

## Author summary
### Why was this study done?

- The relationship between multimorbidity, particularly patterns involving cognitive conditions, and healthcare utilisation, as well as the role of health insurance, remains poorly understood.

### What did the researchers do and find?

- We analysed harmonised data from ageing cohort studies across 31 countries to examine how different multimorbidity patterns were associated with outpatient visits and hospital admissions.

- The relationship between multimorbidity and healthcare utilisation varied across socioeconomic groups, showing a clear socioeconomic gradient.

- Health insurance appeared to buffer the association between multimorbidity and healthcare utilisation.

**What do these findings mean?**

- Patterns of multimorbidity that include cognitive disorders may be linked to lower recorded healthcare utilisation, suggesting potential barriers to accessing care or greater reliance on informal support.

- Health insurance may help reduce socioeconomic inequalities in access to healthcare services among people with multimorbidity.

- Differences in healthcare systems and measurement across countries limit direct comparability of utilisation estimates.

## Introduction

Physical, psychological, and cognitive multimorbidity pose substantial challenges to individuals, healthcare systems and societies. This growing burden is further amplified by global population ageing and shifts in health behaviours, which have contributed to the increasing prevalence of multimorbidity [1–3]. Studies have reported that the overall global prevalence of multimorbidity is 34.9%–39.4%, making multimorbidity a major global health concern whose impact is expected to intensify in the coming decades [4]. Moreover, the complexity of multimorbidity extends beyond physical conditions to frequently involve psychological and cognitive disorders, which complicate management and further strains healthcare systems.

While the intersection of physical and psychological multimorbidity has been explored in previous studies [5,6], few studies have examined cognitive disorders as part of the multimorbidity spectrum [7,8]. Most existing research focuses on physical and psychological multimorbidity and their contribution to the risk of developing cognitive impairment or dementia [9,10], or examines its association with healthcare utilisation [5,11]. However, these studies have not fully assessed the consequences of cognitive-related multimorbidity, which hinders a comprehensive understanding of its impact on healthcare utilisation patterns, particularly given the known complexity of healthcare utilisation in those with cognitive impairment even without multimorbidity [12]. The inclusion of cognitive disorders also reveals more pronounced socioeconomic inequalities in multimorbidity prevalence, particularly in low- and middle-income countries [7]. In this study, we conceptualised multimorbidity as the coexistence of two or more chronic conditions across three domains—physical, psychological, and cognitive (see Methods for details). However, an important gap remains: no studies have fully described the relationship between healthcare use and the interplay of physical, psychological, and cognitive multimorbidity, nor have they thoroughly examined the role of socioeconomic status (SES).

Health insurance plays a crucial role and service as a key mechanism for advancing Universal Health Coverage (UHC). Individuals with multimorbidity experience significant treatment and illness burdens, which often result in increased healthcare utilisation and costs [11]. They also frequently face unmet healthcare needs and inadequate access to appropriate care, particularly in systems that prioritise single disease management over the complex care coordination required for their multifaceted health needs [13,14]. As a key instrument of financial risk protection, health insurance has the potential to mitigate the economic burden of multimorbidity management. Existing research on the impact of health insurance on healthcare utilisation among individuals with multimorbidity is limited. Most studies focus on single-country settings and define multimorbidity narrowly as physical conditions alone or combinations of physical and mental disorders [15,16]. Given ongoing trends of population ageing and the increased vulnerability of patients with cognitive disorders to unmet healthcare needs [17,18], it is crucial to consider the complexities introduced by cognitive disorders in multimorbidity patterns. A clearer understanding of these associations could inform targeted interventions and policies aimed at reducing disparities in healthcare access among individuals with multimorbidity.

Utilising panel data from five longitudinal studies across 31 countries, this study examines the association between healthcare utilisation and physical, psychological, and cognitive multimorbidity, as well as the modifiable role of health insurance in this context, particularly concerning SES. We tested the following hypotheses:

(1) a higher complexity of multimorbid conditions is associated with increased healthcare utilisation; and

(2) these associations would vary across levels of health insurance coverage and SES.

## Methods

### Study design and data sources

This multicohort study used data from six population-based longitudinal studies across 31 countries: Health and Retirement Longitudinal Study (HRS), the China Health and Retirement Longitudinal Study (CHARLS), the Japanese Study of Aging and Retirement (JSTAR), the Korean Longitudinal Study of Aging (KLoSA), the Mexican Health and Aging Study (MHAS) and the Survey of Health, Ageing and Retirement in Europe (SHARE) [19–23]. These studies use broadly similar survey protocols, allowing for coordinated cross-national comparisons. All studies adhere to a biennial design and use standardised measures of economic status, lifestyle, and health; however, residual differences in healthcare system organisation and measurement across cohorts remain. Data were drawn from waves 9–15 of HRS (2008–2020), waves 1–4 of CHARLS (2011–2018), waves 1–3 of JSTAR (2007–2011), waves 1–8 of KLoSA (2006–2020), waves 1–5 of MHAS (2001–2018) and waves 4–9 of SHARE (2008–2019).

All studies received ethical approval, and participants provided written informed consent.

### Sample construction

For this study, we included data from participants aged 50 years and older to ensure comparability across cohorts.

### Measure

**(1) Multimorbidity.** In this study, multimorbidity was defined as the co-occurrence of two or more disorders from any combination of physical, psychological, or cognitive domains, identified during each survey wave of the survey [7]. Participants were classified as having a physical disorder if they self-reported at least one of the following seven chronic conditions: hypertension, diabetes, cancer, lung disease, heart disease, stroke, or arthritis. The presence of a psychological disorder was determined using study-specific psychological assessments (e.g., the Center for Epidemiologic Studies Depression scale), using the same cut-off values as in previous studies (Table A in S1 Appendix). Potential cognitive disorders were assessed using study-specific cognitive tests, with cut-off values either adopted from previous research or determined empirically (Table A in S1 Appendix). For each domain, participants were classified as having a disorder if at least one condition within that domain was present. In this study, the complexity of multimorbidity was conceptualised as the number of morbidity domains—physical, psychological, and cognitive—in which at least one condition was present. Psychological and cognitive disorders were identified using screening instruments with cohort-specific cut-off values rather than clinical diagnoses.

**(2) Outcome.** Healthcare utilisation was measured using two indicators: outpatient visits and inpatient admissions. We aligned recall periods across datasets to enhance comparability, while acknowledging that outpatient and inpatient utilisation may not be fully equivalent across healthcare systems and survey instruments. We categorised healthcare use as binary, indicating whether the participant had outpatient visits or inpatient admissions. To assess the intensity of healthcare use, we additionally calculated the number of visits. Due to recall periods differed across cohorts, utilisation measures were standardised prior to analysis. Outpatient utilisation was expressed as the number of outpatient visits

in the past year, while inpatient utilisation was expressed as the number of hospital admissions in the past two years. Detailed definitions and measurement of variables are provided in Table B in S1 Appendix.

**(3) Health insurance.** Given the differences in health insurance systems across countries, we treated health insurance at the individual level as a binary variable (covered versus not covered), encompassing public, private, or any other form of coverage. This captures the presence or absence of financial protection but does not imply cross-country equivalence of benefit generosity, access conditions, or service scope. Detailed definitions and measurement of this variable are provided in Table C in S1 Appendix.

**(4) Covariates.** In each study, covariates were assessed at the selected survey wave. These included basic demographic characteristics: age, gender (male versus female), educational attainment (middle school or below, high school, and college or above), work status (unemployed versus employed), and marital status (not married versus married). SES was measured using household wealth, which was categorised into within-country quartiles (Q1–Q4), recalculated at each wave to reflect temporal shifts in wealth distribution. Additionally, lifestyle factors—BMI, alcohol use, tobacco use, and physical activity—were also included as covariates in all regression models because they may confound the association between multimorbidity and healthcare utilisation. BMI was calculated as weight (kg) divided by height squared (m$^2$) and categorised into underweight (<18.5 kg/m$^2$), normal weight (18.5–24.9 kg/m$^2$), and overweight and obesity (≥25 kg/m$^2$). All covariates were measured contemporaneously at each wave. Covariate adjustment was intended to improve comparability across groups and describe associations. Definitions and measurement methods for alcohol use, tobacco use, and physical activity are provided in Table D in S1 Appendix.

Fig A in S1 Appendix presents the directed acyclic graph

## Statistical analysis

First, we conducted descriptive analyses of the prevalence of multimorbidity, stratified by country and age group. We then examined the probability of healthcare utilisation across different multimorbidity patterns, focussing on outpatient and inpatient services. These analyses were further stratified by SES to provide additional insight into socioeconomic differences in healthcare use. To assess the associations between multimorbidity and healthcare utilisation, we used longitudinal random-effects logistic regression with individual-level random intercepts to account for within-individual correlation over time. This allowed us to estimate the likelihood of outpatient visits and inpatient admissions in relation to multimorbidity patterns. We additionally estimated marginal models and computed average marginal effects (Fig E in S1 Appendix).

To analyse count outcomes (number of outpatient visits; number of inpatient admissions), we used random-effects negative binomial models. Overdispersion was formally assessed using the dispersion parameter from these models, and the estimated dispersion component ($r$) was consistently greater than zero with 95% confidence intervals excluding zero across countries, indicating substantial extra-poisson variation and justifying the use of negative binomial regression (Tables I and K in S1 Appendix). Specifically, we estimated country-specific models to obtain within-country associations. These results were then summarised into country-specific estimates using the random-effects meta-analysis. Given substantial between-country heterogeneity, pooled meta-analytic estimates are presented as descriptive summaries of the distribution of associations across countries. To further explore potential sources of between-country heterogeneity, we conducted exploratory meta-regression analyses using country-level indicators, including the UHC index, GDP per capita, income inequality, and health expenditure.

To examine the differential effects across population subgroups, we performed subgroup analyses stratified by SES and health insurance coverage. These analyses followed the same regression approach but excluded the stratification variable. For these models, data from all countries were pooled, with country fixed effects included to account for time-invariant country-level differences in healthcare systems and survey context.

For the logistic regression analyses, we reported associations as odds ratios (ORs) adjusted for relevant covariates, with 95% confidence intervals (CIs). For the random-effects negative binomial regression analysis, we presented incident

rate ratios (IRRs) adjusted for covariates, also with 95% CIs. No sample weights were applied in our analysis due to the multi-cohort design and inconsistent weighting procedures across studies. Standard errors were adjusted at the individual level. Statistical significance was defined as a two-sided P value <0.05. All analyses were performed using Stata (version 15.0). The multiple imputation (with 20 sets) was employed to handle the missing values.

The data were publicly available. The use of public, secondary, de-identified data made the present study exempt from institutional research ethics review. This study is reported as per STROBE guideline (S1 Checklist).

## Results

The average age of respondents across countries ranged from 62.8 to 70.2 years, with a slightly higher proportion of females, ranging from 50.7% to 62.8%.

In terms of multimorbidity patterns, physical-psychological multimorbidity showed the highest prevalence, ranging from 7.5% in China to 32.6% in Lithuania. The prevalence of physical- psychological-cognitive multimorbidity ranged from 0.4% in Finland to 17.6% in South Korea.

Regarding healthcare utilisation, outpatient care use varied widely. In China (CHARLS), where the recall period was one month, outpatient utilisation was lowest at 18.9%. In the other cohorts, measured over 12 months (24 months in KLoSA), Germany reported the highest outpatient utilisation at 94.4%. The median outpatient visit within one year was 3 (IQR 1–7.5). For inpatient care, the pooled utilisation proportion was 14.5%.

The complexity of multimorbidity patterns increased with age, with slight gender differences (Fig 1). As age increased, the proportion of individuals with physical-psychological-cognitive multimorbidity rose, from 1.8% among individuals aged 50–59 years to 17.4% among those aged 90 years or older.

In men, the age-related pattern of multimorbidity was comparable to that observed in women.

An analysis of the relationship between multimorbidity patterns and healthcare utilisation (Fig 2) revealed several notable findings. For outpatient care, individuals with cognitive disorders alone had the lowest utilisation proportion at 34.0%, while those with only physical conditions had the highest utilisation proportion at 88.0%. Those with the most complex multimorbidity pattern—physical-psychological-cognitive multimorbidity—had an outpatient care utilisation proportion of 82.1%.

For inpatient services, utilisation increased with greater multimorbidity complexity. Individuals with physical-psychological multimorbidity had the highest hospitalisation proportion at 23.0%, followed closely by those with physical-psychological-cognitive multimorbidity at 22.8% (Fig 2).

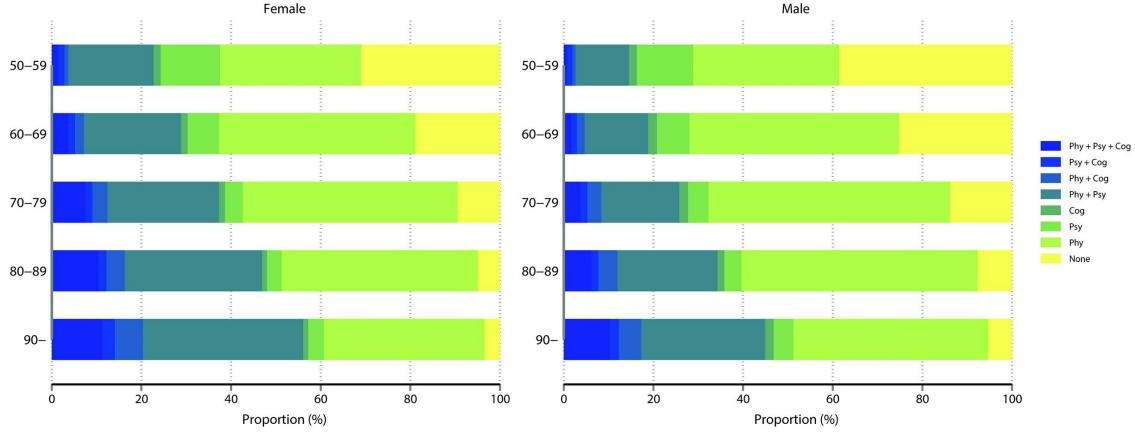

**Fig 1. Prevalence of physical, psychological, and cognitive multimorbidity by gender and age.**

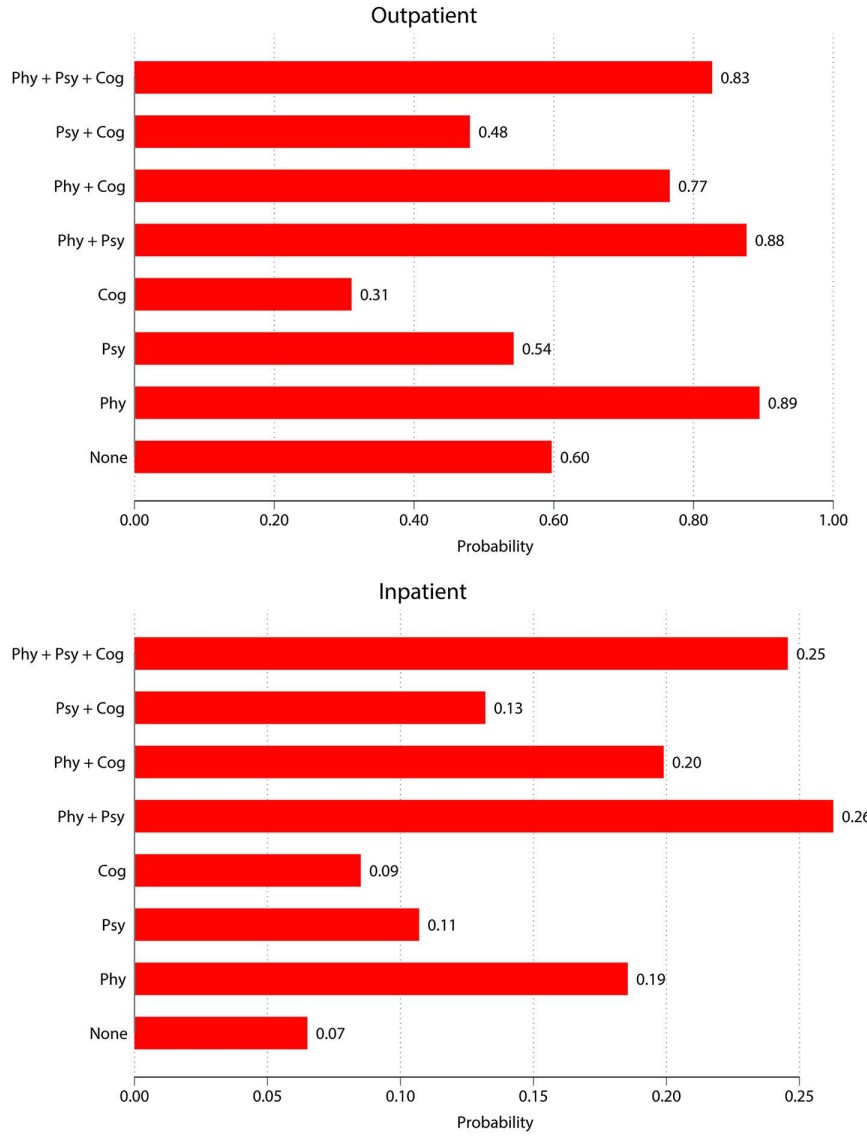

**Fig 2. Probability of healthcare utilisation across different multimorbidity patterns.**

## The association between multimorbidity and healthcare utilisation

The adjusted association between multimorbidity patterns and outpatient care utilisation reveals a complex pattern (Fig 3). Compared with individuals without any conditions, those with cognitive disorders alone had significantly lower outpatient utilisation (OR 0.62, 95% CI [0.48, 0.76]; $p < 0.001$). Unexpectedly, individuals with the most complex multimorbidity pattern—physical-psychological-cognitive multimorbidity—did not exhibit the highest outpatient care utilisation (OR 3.21, 95% CI [2.39, 4.03]; $p < 0.001$), and had lower odds than those with a single physical condition (OR 5.28, 95% CI [4.56, 6.00]; $p < 0.001$). The highest outpatient utilisation was observed among those with physical-psychological multimorbidity, representing a 7.84-fold (OR 7.84, 95% CI [6.59, 9.10]; $p < 0.001$) increase compared with those without any conditions.

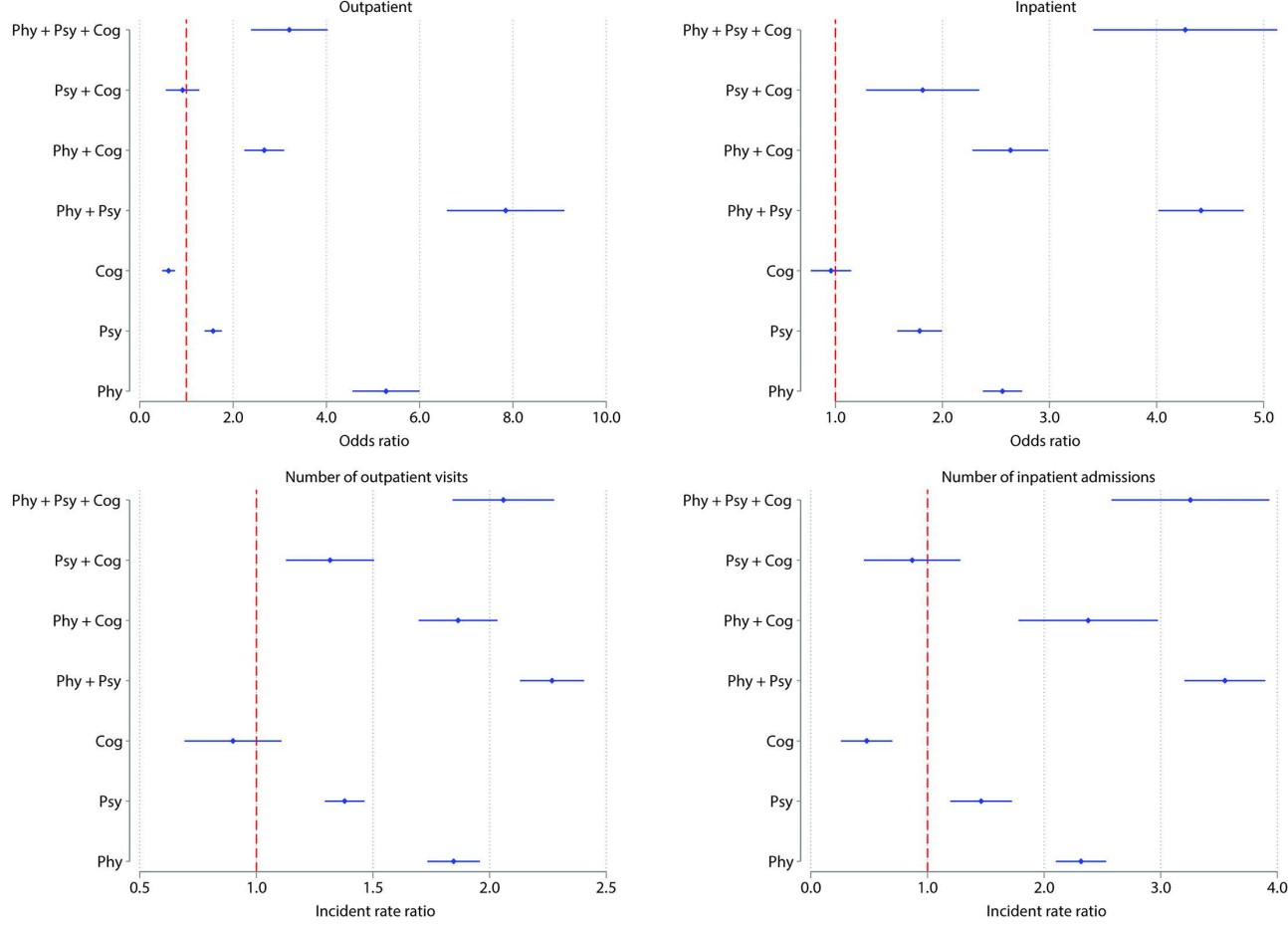

**Fig 3. The association between healthcare utilisation and multimorbidity (result of meta-analysis).**

However, when analysing outpatient visit number rather than binary utilisation, similar patterns were observed, although effect sizes were smaller. The most complex multimorbidity pattern—physical-psychological-cognitive multimorbidity—was not associated with the highest visit frequency (IRR 2.06, 95% CI [1.84, 2.28]; $p < 0.001$).

For inpatient care, a positive association was found between the complexity of multimorbidity patterns and hospitalisation. However, adding a cognitive disorder to an existing physical or psychological condition was not associated with higher inpatient utilisation. Similarly, for number of inpatient admissions, the pattern mirrored the trend seen with the binary inpatient outcome. Individuals with cognitive disorders had lower inpatient service utilisation compared with those without any conditions (IRR 0.48, 95% CI [0.26, 0.70]; $p = 0.004$). Among all multimorbidity patterns, individuals with physical-psychological multimorbidity had the highest inpatient utilisation (IRR 3.55, 95% CI [3.20, 3.90]; $p < 0.001$). We additionally estimated marginal predicted counts to translate relative measures into absolute differences (Table G in S1 Appendix). Moreover, the associations between multimorbidity and healthcare utilisation showed a subtle gradient in relation to SES, particularly in outpatient care, where the association weakened as SES increased (Fig 4). We tested the multimorbidity × SES interaction, and the results are presented in Table H in S1 Appendix. Country-specific estimates are provided in Tables I–L in S1 Appendix, and details of the meta-analysis are presented in Table M in S1 Appendix.

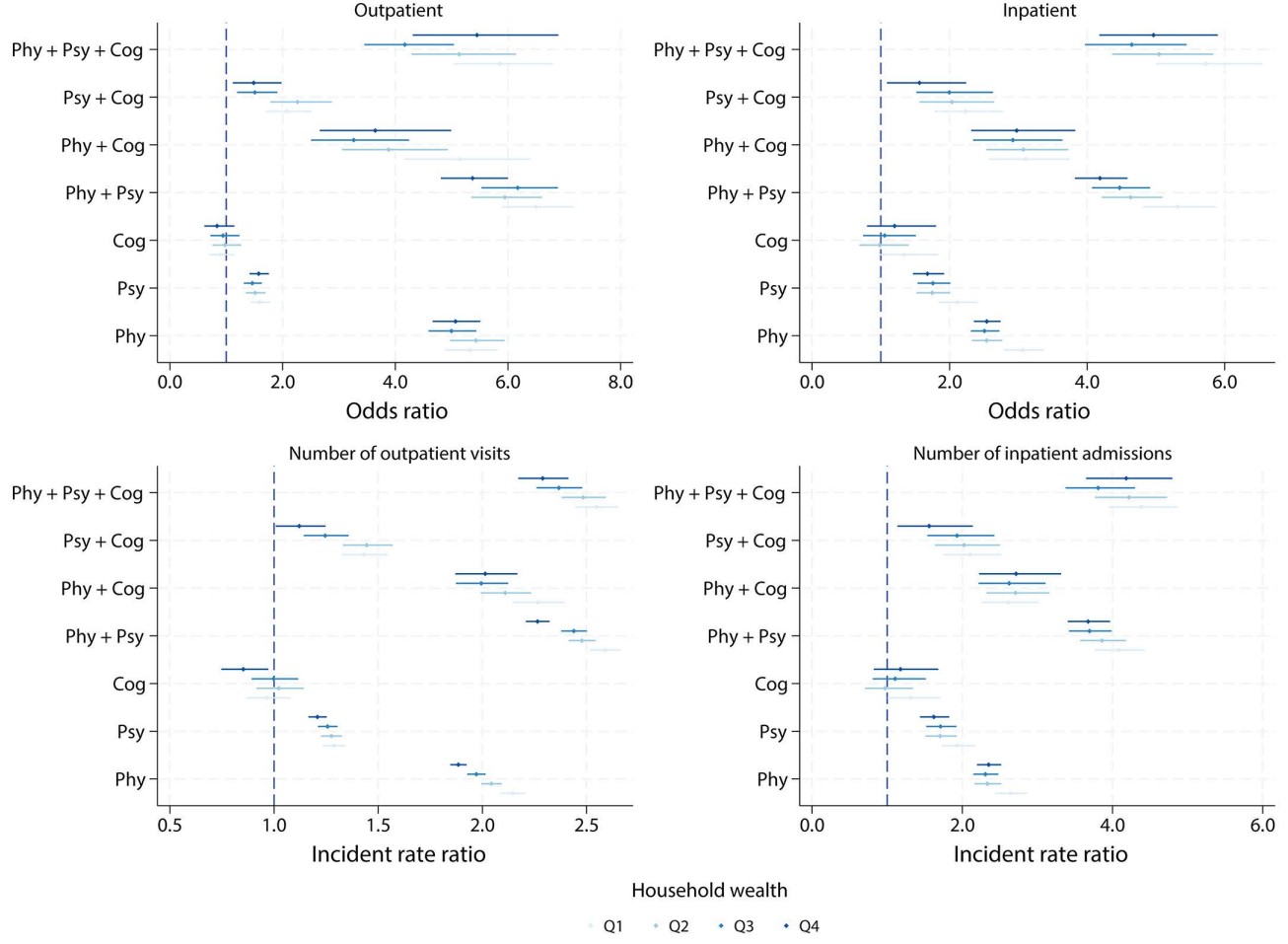

**Fig 4. The association between healthcare utilisation and multimorbidity (grouped by SES).**

## The role of health insurance

Stratified analyses by health insurance status suggested different patterns in the relationship between multimorbidity and healthcare use, especially for outpatient care (Fig 5). Specifically, among individuals with health insurance coverage, the association between multimorbidity and outpatient care use was generally weaker. This was particularly evident for those with physical-psychological-cognitive multimorbidity (without insurance: OR 6.22, 95% CI [5.33, 7.25]; $p < 0.001$; with insurance: OR 3.40, 95% CI [3.03, 3.82]; $p < 0.001$), physical-psychological multimorbidity (without insurance: OR 8.32, 95% CI [7.76, 8.93]; $p < 0.001$; with insurance: OR 3.01, 95% CI [2.76, 3.29]; $p < 0.001$), and single physical conditions (without insurance: OR 5.67, 95% CI [5.39, 5.97]; $p < 0.001$; with insurance: OR 3.53, 95% CI [3.23, 3.85]; $p < 0.001$). Similar patterns were observed for inpatient care utilisation, although the differences were less pronounced compared with outpatient care. To further assess the statistical significance, we included a multimorbidity × health insurance interaction term in the models. Results are presented in Table N in S1 Appendix.

Further analyses comparing ORs and IRRs between insured and uninsured individuals across SES quartiles for each multimorbidity pattern showed that, for outpatient care—both binary utilisation and visit frequency—most differences were positive (Fig C in S1 Appendix). This pattern was more pronounced in lower SES groups and was consistent with

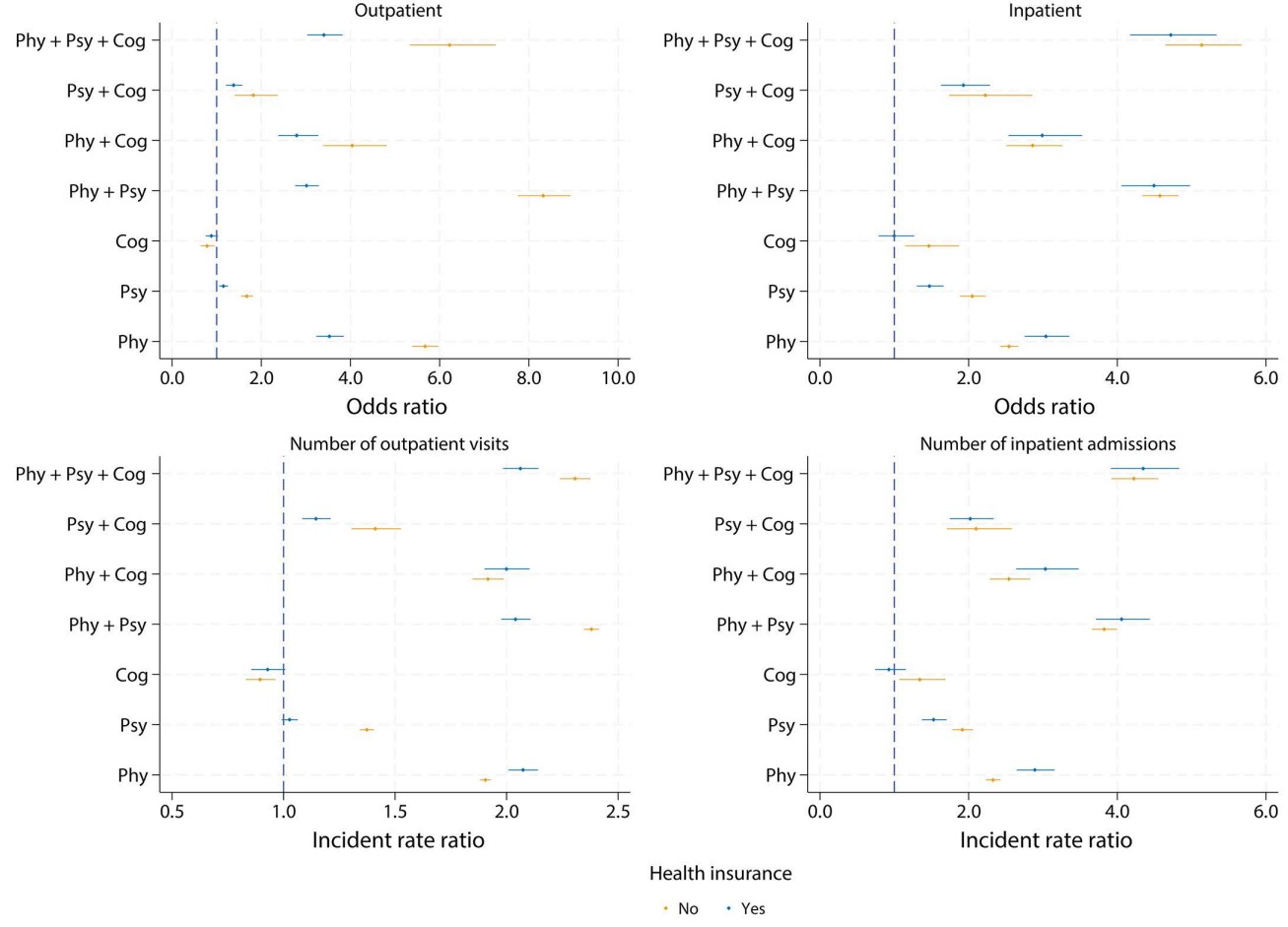

**Fig 5. The association between healthcare utilisation and multimorbidity (grouped by health insurance).**

a negative socioeconomic gradient in the association between health insurance and outpatient utilisation. However, for inpatient care, the relationship with SES showed the opposite pattern. These analyses were exploratory and were not interpreted as definitive evidence of effect modification.

## Sensitivity analyses

We conducted multiple sensitivity analyses to assess the robustness of our findings. First, to address potential measurement unit discrepancies, we reanalysed our data. For outpatient care, we excluded CHARLS, HRS and KLoSA, and for inpatient care, we excluded HRS and KLoSA (Fig D in S1 Appendix), yielding results consistent with our primary findings. Recall periods differed across surveys: outpatient care was measured over 2 years in HRS, 1 month in CHARLS, and 2 years in KLoSA, versus 1 year in other countries; inpatient care was measured over 2 years in HRS and KLoSA and 1 year in other countries. Second, we estimated random-effects Poisson regression models for the number of healthcare visits (Fig D in S1 Appendix) and calculated marginal effects derived random-effects probit models for the binary outpatient and inpatient care (Fig E in S1 Appendix), given the high prevalence of outpatient use in several cohorts. Results remained stable across model specifications. Third, we tested an alternative SES by replacing household wealth with education attainment in stratified analyses and observed similar trends (Fig F in S1 Appendix). In addition, we re-analysed

our results treating SES as an ordinal variable (Fig G in S1 Appendix). Fourth, to account for the hierarchical data structure, we re-estimated the main models using multilevel models (random intercepts and slopes models) and reported both weighted and unweighted results, which remained consistent with the main findings (Table S in S1 Appendix; Figs H–J in S1 Appendix). Fifth, to examine potential model misspecification related to age structure, we restricted the analysis to adults aged 65 years and above and re-estimated the models, again observing stable results (Fig K in S1 Appendix). Sixth, complete-case analyses were conducted, yielding results consistent with the multiply imputed analyses (Fig L in S1 Appendix). Seventh, to further investigate between-country heterogeneity, we conducted exploratory meta-regression analyses using country-level indicators. Compared with univariable random-effects meta-analysis, these models showed a general tendency toward lower residual heterogeneity ($I^2$_res) across most multimorbidity categories. However, substantial heterogeneity remained, reductions varied by outcome and multimorbidity pattern (Tables T–W in S1 Appendix). Eighth, to address concerns regarding temporal ordering and potential adjustment for time-varying covariates measured contemporaneously with the outcome, we re-estimated the models using only baseline covariates. The results were consistent in direction and magnitude with the primary analyses (Figs M in S1 Appendix). Ninth, we estimated hurdle models in which the first component modelled the probability of any utilisation using logistic regression and the second part modelled the visits number using truncated negative binomial regression. Results were directionally consistent with the primary analyses (Figs N in S1 Appendix). Furthermore, model adequacy was further assessed using comparative fit statistics (AIC/BIC) (Table X in S1 Appendix). Tenth, to assess potential exposure misclassification of cognitive and psychological disorder, we re-estimated all models using alternative cut-offs for multimorbidity classification. Results remained robust (Figs O and P in S1 Appendix). Finally, to assess potential bias due to differential attrition, we examined follow-up proportions across waves (Table E in S1 Appendix) and estimated logistic regression models predicting loss to follow-up (Table Y in S1 Appendix). No strong predictors of attrition were identified.

## Discussion

This study demonstrates a significant but "non-monotonic" relationship between healthcare utilisation and multimorbidity using cross-national data. Specifically, the presence of cognitive disorders was associated with a reduced likelihood of outpatient service use in individuals with co-occurring conditions. Additionally, this relationship between multimorbidity and healthcare use demonstrated a socioeconomic gradient. Furthermore, we found that health insurance exerts a buffering effect, attenuating some of the SES-related differences in healthcare use and potentially access to care.

The multimorbidity pattern that included cognitive disorders was associated with lower healthcare utilisation compared with physical-only multimorbidity. Unlike previous studies, which have generally reported that healthcare utilisation increases with the number of chronic conditions or the complexity of multimorbidity patterns [3,11,24–26]—primarily focussing on physical or mental multimorbidity, our findings suggest a notable reduction in healthcare use among individuals with cognitive disorder-related multimorbidity. These findings highlight the complexity of healthcare utilisation patterns as multimorbidity burden evolves and challenge the assumption that greater complexity necessarily results in higher recorded service use.

Two possible explanations may account for these findings. First, from the demand-side perspective, individuals with cognitive disorders may receive much informal care from family or social networks, which is not captured by conventional measures of healthcare utilisation [17,27,28]. In disorder-specific analyses (Tables O–R in S1 Appendix), cognitive disorders were consistently associated with reduced outpatient service access. Although this negative association was attenuated when cognitive disorders co-occurred with other conditions, individuals with cognitive-related multimorbidity continued to exhibit lower utilisation relative to other multimorbidity patterns. In addition, cognitive disorders in our study are identified through screening assessments rather than clinical diagnoses. Differences in screening tools across cohorts may lead to under-recognition or misclassification of cognitive impairment, particularly among individuals with mild symptoms, which could reduce the likelihood that healthcare needs are identified and documented. While direct

evidence is limited, this interpretation aligns with previous research on cognitive disorders that demonstrates high levels of unmet healthcare needs in such populations [17]. Second, from the supply-side perspective, we observed differing results between outpatient care (binary) and the number of outpatient visits (count), offering an alternative explanation. Binary outpatient utilisation reflects access to care, while visit counts reflect the intensity of use. The absence of a clear increase in the likelihood of obtaining any outpatient care as multimorbidity complexity increases may reflect persistent system-level barriers, including limited availability of dementia-related services, fragmented care pathways, and insufficient integration between medical and social care [29]. However, once individuals access care, their complex and multiple needs may lead to greater service intensity, as reflected in visit frequency. These findings—particularly the association between cognitive disorder-related multimorbidity and lower healthcare use—differ from some previous studies [30–32].

The relationship between multimorbidity and healthcare utilisation indicates significant SES-related inequalities, with associations appearing stronger among individuals with lower SES. It should be noted that these gradient patterns reflect relative differences within countries rather than absolute differences across nations. This may be because individuals with higher SES generally have greater capacity to access and navigate healthcare resources [33]. This finding extends previous research, which had documented SES inequalities in the prevalence of physical-psychological-cognitive multimorbidity [7,8]. It suggests that disadvantaged groups may experience compounded challenges related to both illness burden and the demands of managing multiple conditions.

Health insurance potentially plays a key mechanism for narrowing disparities in healthcare utilisation, particularly those related to SES. As a core component of UHC, health insurance is designed to protect individuals and families from financial hardship due to high healthcare expenditures [16]. In this study, the association between multimorbidity and healthcare use was less pronounced among insured individuals than among those without coverage, especially regarding outpatient services. These findings suggest that insurance coverage may attenuate disparities in healthcare utilisation related to multimorbidity, potentially lowering the unmet healthcare needs. However, this buffering effect was less evident for inpatient services. This may reflect inherent characteristics of multimorbidity and the relatively lower elasticity of demand for hospital care compared with outpatient care [34]. Chronic conditions typically require ongoing management, making outpatient services more sensitive to financial and access-related barriers. Further analyses suggest that the protective association of health insurance with outpatient care was more pronounced among individuals with lower SES, indicating a potential moderating role of insurance in reducing utilisation gaps among disadvantaged populations. However, formal insurance coverage does not necessarily translate into effective financial protection. This finding contributes to understanding health insurance in promoting equitable access to necessary health services for vulnerable populations. Therefore, this study enhances our understanding of the role of individual-level insurance coverage in shaping healthcare utilisation among people with multimorbidity [15].

Our study provides new evidence on physical, psychological, and cognitive multimorbidity and their associations with healthcare utilisation, and how these relationships are moderated by SES and health insurance, with several policy implications. First, the complex and non-linear utilisation patterns observed suggest that current healthcare structures may not adequately recognise or respond to the multidimensional needs of individuals with multimorbidity. Our findings demonstrate that the inclusion of cognitive disorders in multimorbidity patterns introduces additional complexity in healthcare needs, rather than a straightforward or monotonic increase. Although our study does not directly evaluate care models, these findings align with broader evidence indicating that more coordinated and person-centred approaches—including improved integration between healthcare and family or social support—may help address gaps in identification, management, and support for this population. Current clinical guidelines are insufficient in meeting the complex needs of patients with multimorbidity [3,29]. Secondly, expanding health insurance coverage may play an important role in promoting health equity among individuals with multimorbidity. As multimorbidity becomes increasingly prevalent and complex, placing greater demands on healthcare systems, progress toward achieving UHC will face significant challenges. Although health insurance has shown buffering effects, sustained impact will likely require reforms in both financing and service delivery

systems to better address multimorbidity beyond traditional single-disease models. Lastly, our study highlights the urgent need to address SES inequalities in healthcare services. Equity-oriented policies and programmes are essential to reduce social disparities in healthcare access for multimorbidity. Such efforts may require targeted allocation of health resources and interventions to disadvantaged groups.

This study has two notable strengths. First, the panel data analysis uses cross-national data to examine healthcare utilisation differences among individuals with physical, psychological, and cognitive multimorbidity, addressing limitations in prior studies—such as challenges related to cross-cultural generalizability, statistical conclusion validity, and measurement validity. Second, the study offers a detailed and nuanced analysis of the role of health insurance, exploring not only its buffering effects but also its interaction with socioeconomic inequalities in healthcare use.

This study had several limitations that should be acknowledged. First, the information on physical, psychological, and cognitive disorders, as well as healthcare use, was self-reported, which may have introduced recall bias and led to underestimation [11]. Second, the measurement of psychological and cognitive disorders may have varied across datasets. We used cohort-specific cut-offs for psychological and cognitive assessments, which could have introduced exposure misclassification and contributed to between-country heterogeneity. In addition, screening-based definitions may not be fully comparable to clinically diagnosed disorders. However, this approach is widely used and considered appropriate for cross-cohort comparisons [7,8]. Third, despite efforts to align recall periods for healthcare utilisation, residual non-comparability of outpatient and inpatient care definitions across cohorts remains. Surveys differ in the extent to which they distinguish primary versus specialist care or public versus private services. Sensitivity analyses restricted to countries with identical recall periods yielded consistent findings. Fourth, substantial between-country heterogeneity was observed for several outcomes. Although exploratory meta-regression analyses explained part of this variation, unmeasured structural and health system factors may remain. Fifth, given the study context and design, this study cannot establish a causal relationship between multimorbidity and healthcare utilisation. Sixth, dichotomising health insurance status may mask important cross-national differences. Future research should investigate how different health financing systems shape the relationship. Seventh, given the multiple comparisons performed in subgroup and interaction analyses, findings should be interpreted cautiously in light of the potential for type I error inflation. Finally, due to data limitations, we were unable to account for supply-side factors. The analyses adjusted only for individual-level characteristics and did not consider healthcare system-related factors such as the type of health insurance scheme and physician access, which might also influence healthcare use.

Rapid population ageing underscores the need to better understand the complexities of multimorbidity patterns, particularly those involving cognitive disorders. These complexities significantly affect healthcare use behaviour and highlight the diverse healthcare needs. Addressing these challenges will require movement toward more person-centred and coordinated care models that integrate healthcare services with family and social support systems. Furthermore, strengthening multimorbidity-responsive health insurance policies to promote equity and accessibility in healthcare services is essential. Such efforts are vital not only for improving individual health outcomes but also for advancing progress toward UHC and Sustainable Development Goals.

## Supporting information

**S1 Appendix. Table A:** The definition of physical, psychological and cognitive disorder. **Table B:** The definition of healthcare use. **Table C:** The definition of health insurance. **Table D:** The definition of lifestyles. **Table E:** Basic demographics by study. **Table F:** Basic demographics by country. **Table G:** Marginal predicted counts. **Table H:** The statistical test results of multimorbidity*SES. **Table I:** The association between multimorbidity and outpatient by country. **Table J:** The association between multimorbidity and number of outpatient visits by country. **Table K:** The association between multimorbidity and inpatient by country. **Table L:** The association between multimorbidity and number of inpatient admissions by country. **Table M:** The meta-analysis details. **Table N:** The statistical test results of multimorbidity*health insurance.

**Table O:** Association between each disorder domain and outpatient care by country. **Table P:** Association between each disorder domain and inpatient care by country. **Table Q:** Association between each disorder domain and number of outpatient visits by country. **Table R:** Association between each disorder domain and number of inpatient admissions by country. **Table S:** Variance components and ICC from mixed models. **Table T:** Changes in between-country heterogeneity ($I^2$) (outpatient). **Table U:** Changes in between-country heterogeneity ($I^2$) (inpatient). **Table V:** Changes in between-country heterogeneity ($I^2$) (number of outpatient visits). **Table W:** Changes in between-country heterogeneity ($I^2$) (number of inpatient admissions). **Table X:** Model fit comparison across models. **Table Y:** Predictors of attrition (logistic regression models). **Fig A:** Directed acyclic graph. **Fig B:** Prevalence of physical, psychological, and cognitive multimorbidity by country. **Fig C:** The OR and IRR differences between individuals with and without health insurance (grouped by multimorbidity and SES). **Fig D:** The association between healthcare utilisation and multimorbidity (specific cohorts and poisson model analyses). **Fig E:** The association between binary healthcare utilisation and multimorbidity (marginal effects derived from probit model). **Fig F:** The association between healthcare utilisation and multimorbidity (grouped by education attainment). **Fig G:** The association between healthcare utilisation and multimorbidity (treating SES as an ordinal variable). **Fig H:** The association between healthcare utilisation and multimorbidity (modelled by random intercepts model). **Fig I:** The association between healthcare utilisation and multimorbidity (modelled by weighted multilevel model). **Fig J:** The association between healthcare utilisation and multimorbidity (modelled by random slopes model). **Fig K:** The association between healthcare utilisation and multimorbidity (aged 65 and above). **Fig L:** The association between healthcare utilisation and multimorbidity (complete-case). **Fig M:** The association between healthcare utilisation and multimorbidity (only control for baseline covariates). **Fig N:** The association between healthcare utilisation and multimorbidity (hurdle model). **Fig O:** The association between healthcare utilisation and multimorbidity (stricter screening cut-offs). **Fig P:** The association between healthcare utilisation and multimorbidity (more lenient screening cut-offs).
(DOCX)

**S1 Checklist. The STROBE checklist is best used in conjunction with this article (freely available on the websites of PLoS Medicine at http://www.plosmedicine.org/, Annals of Internal Medicine at http://www.annals.org/, and Epidemiology at http://www.epidem.com/).** Information on the STROBE Initiative is available at http://www.strobe-statement.org.
(DOCX)

**S1 Data. Minimal dataset.**
(XLSX)

## Acknowledgments

The authors gratefully acknowledge the support from the study participants involved in study.

## Author contributions

**Formal analysis:** Yanshang Wang.

**Funding acquisition:** Ping He.

**Investigation:** Yanshang Wang, Zhenyu Shi, Ping He.

**Methodology:** Yanshang Wang, Chang Cai, Alex Bottle, Mansour Taghavi Azar Sharabiani.

**Supervision:** Benedict Hayhoe, Ping He.

**Visualization:** Yanshang Wang.

**Writing – original draft:** Yanshang Wang.

**Writing – review & editing:** Yanshang Wang, Chang Cai, Zhenyu Shi, Qian Gao, Alex Bottle, Mansour Taghavi Azar Sharabiani, Joshua Stott, Benedict Hayhoe, Ping He.

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
