## [Editor Report · Decision Letter 0]

12 Dec 2025

Dear Dr He,

Thank you for submitting your manuscript entitled "Multimorbidity, health service use, and health insurance by socioeconomic groups in 31 countries: a multi-cohort study" for consideration by PLOS Medicine.

Your manuscript has now been evaluated by the PLOS Medicine editorial staff, and I am writing to let you know that we would like to send your submission out for external peer review.

In the revised manuscript file please provide response to AE comments:

1. Supplementary, country-specific analyses to evaluate whether—and how—the findings vary across settings.

2. Another concern relates to the low rates of service use for cognitive disorders. One possible explanation is that many health systems offer limited support or treatment options for people with dementia. Therefore, lower utilization in the absence of physical or mental health comorbidities may not be unexpected. Nevertheless, it would be helpful to examine the associations between each of the three disorders and service utilization within each country to better interpret and contextualize the results.

I have put a deadline for submission of revised manuscript by the end of next week. Please, let me know if you need more time to fulfil AEs requests. Once we receive revised manuscript I will consult with AE if the changes are satisfactory and decide next steps.

Please, have in mind that between 20th December and 5th of January Editorial Team is Out of office so there will be delay in decision process and responses regarding manuscript.

For clinical studies, please upload a copy of your trial study protocol as a supporting information file. The study protocol should be the version submitted for approval to the institutional review board or ethics committee, should include any amendments to the study protocol, as well as the date of their approval by the institutional review or ethics committee. Please also detail any deviations from the study protocol in the Methods section of your manuscript. The editors will consider the protocol and study conduct prior to a final decision for external review.

Please re-submit your manuscript within 6 days, i.e. by Dec 19 2025 11:59PM.

Kind regards,

Katarzyna Starowicz, PhD

Associate Editor

PLOS Medicine

kstarowicz@plos.org

---

## [Decision Letter · Decision Letter 1]

3 Feb 2026

Dear Dr He,

Many thanks for submitting your manuscript "Multimorbidity, health service use, and health insurance by socioeconomic groups in 31 countries: a multi-cohort study" (PMEDICINE-D-25-04109R1) to PLOS Medicine. The paper has been reviewed by subject experts and a statistician; their comments are included below and can also be accessed here: [LINK]

For this version of your manuscript, we sought reviews from a combination of previous reviewers and new subject-matter expert. As you will see in the reports, the reviewers continue to raise substantial concerns regarding the methodology and the extent to which prior comments have been addressed.

After careful discussion with the editorial team and an academic editor with relevant expertise, we have decided to invite you to revise the manuscript in response to the reviewers’ comments.

Please note that we are considering this revision to be a final round. To be considered for publication, all reviewer concerns must be fully and satisfactorily addressed in your revised submission.

We ask that you submit your revision by Feb 24 2026 11:59PM. However, if this deadline is not feasible, please contact me by email, and we can discuss a suitable alternative.

Don't hesitate to contact me directly with any questions (kstarowicz@plos.org).

Best regards,

Katarzyna

Katarzyna Starowicz, PhD

Associate Editor

PLOS Medicine

kstarowicz@plos.org

Comments from the reviewers:

Reviewer #1: Major Comments

1) Despite extensive sensitivity analyses, the manuscript continues to report and interpret pooled effect estimates in the presence of extremely high between-country heterogeneity (I² frequently >90%). While multilevel models are presented as robustness checks, the manuscript does not sufficiently justify the meaningfulness of pooled odds ratios and IRRs under such conditions.

2) The analytical framework does not adequately address temporal ordering and time-varying confounding. It remains unclear whether key covariates (e.g., insurance status, SES, employment) are modelled contemporaneously or lagged, raising concerns about conditioning on post-exposure variables and potential bias.

3) The manuscript still lacks explicit reporting of model diagnostics, particularly for the random-effects negative binomial models. Sensitivity analyses using alternative model families do not substitute for confirmation that key assumptions (e.g., overdispersion, convergence, adequacy of the random-effects structure) were assessed and met in the primary analyses.

4) Residual non-comparability of healthcare utilisation definitions across cohorts remains a major limitation. Standardising recall periods does not resolve fundamental differences in what constitutes outpatient or inpatient care across studies. This issue remains under-emphasised in the main Methods and should be more prominently acknowledged as a threat to internal validity.

5) The continued use of complete-case analysis as the primary approach, with multiple imputation relegated to sensitivity analyses, remains problematic given the scale and likely non-random nature of missingness across cohorts. The potential for selection bias—particularly by socioeconomic status and health status—has not been sufficiently explored.

6) Cognitive and psychological disorders are identified using screening instruments with cohort-specific cut-offs, yet are treated as error-free binary exposures. No quantitative assessment of exposure misclassification (e.g., alternative cut-offs, exclusion of borderline cases) is provided, despite the centrality of cognitive disorders to the main conclusions.

7) Although variance components are now reported, the manuscript does not discuss whether random slopes for key exposures were considered. Given pronounced cross-national heterogeneity, restricting country-level variation to random intercepts may be overly limiting and could obscure meaningful differences in associations.

Minor Comments

8) The manuscript does not clearly state whether robust or cluster-adjusted standard errors were used in panel models, particularly given repeated observations within individuals and clustering within countries.

9) While interaction terms for SES and health insurance are now included, the implications of multiple testing and limited power for interaction analyses across several outcomes are not sufficiently discussed.

10) The rationale for preferring random-effects logistic regression over marginal models (e.g., GEE), particularly for policy-relevant average effects, is not provided.

11) Although marginal effects are presented in supplementary analyses, the continued emphasis on odds ratios for common outcomes (e.g., outpatient utilisation) risks misinterpretation and should be more clearly caveated in the main text.

12) The potential impact of survey attrition is not discussed. Differential loss to follow-up by multimorbidity status or SES could bias estimates even when random effects are used.

13) The manuscript does not report whether collinearity diagnostics were assessed, despite inclusion of correlated socioeconomic variables (wealth, education, employment, insurance) in the same models.

14) For count outcomes, the authors do not clarify whether zero-inflation was assessed or whether alternative count models (e.g., hurdle or zero-inflated models) were considered, particularly in countries with high proportions of non-users.

Reviewer #2:

Title: Multimorbidity, health service use, and health insurance by socioeconomic groups in 31 countries: a multi-cohort study

Authors: Yanshang Wang; Chang Cai; Zhenyu Shi; Qian Gao; Alex Bottle; Mansour Taghavi Azar Sharabiani; Joshua Stott; Benedict Hayhoe; Ping He

Thank you for the opportunity to review this manuscript. Please see my specific feedback to the authors below. The authors have made a good effort to examine their research question in multi-country, multi-study data. Yet the harmonisation and heterogeneity of the datasets and data sources poses tremendous challenges in these types of studies, which should be noted and discussed carefully throughout the manuscript. The carelessness with the language (and neglecting to address this very point made by a previous reviewer) takes away from conveying the message.

Major comments

1. In Methods and throughout the manuscript, the authors refer to 5 studies being included (China Health and Retirement Longitudinal Study (CHARLS), the Japanese Study of Aging and Retirement (JSTAR), the Korean Longitudinal Study of Aging (KLoSA), the Mexican Health and Aging Study (MHAS) and the Survey of Health, Ageing and Retirement in Europe (SHARE)). What are the additional studies (ELSA and HRS) mentioned in various places throughout the manuscript (e.g. sensitivity analyses and Appendix Tables 4 and 5)? A previous reviewer (#3, comment 2) asked the authors about the same issue.

2. Results and Figures 1-3: how many people had multimorbidity consisting of physical -diseases only or psychological disorders only? There groups seem to have been overlooked in these results. Or, if the groups titled e.g. "psychological disorder" in fact represent people with multiple psychological disorders, how many people had only one psychological, only one physical and only one cognitive disorder?

3. Were dementias included in the cognitive disorders? If not, why? What were the implications of this to the findings? The authors allude to this in the discussion but it would be worth a more detailed note.

4. Methods and throughout the manuscript: the authors will need to specify what outpatient healthcare means in the studies they have used in their analysis. Is this primary or secondary healthcare? Are public healthcare system appointments only included, or do the data include private (and privately paid for) or occupational (paid for by the employer in some European countries) healthcare appointments? What are the implications of including/excluding these data sources?

5. How were the associations of multimorbidity with healthcare utilisation examined in data from countries with universal healthcare - where essentially everyone (bar certain marginal groups, e.g. illegal immigrants, who are unlikely to participate in health surveys) are covered by health insurance?

6. Were ORs and IRRs pooled in meta-analyses? Pooling estimates of cross-sectional estimates (ORs) and incidence estimates (IRRs) does not make sense. Mathematically, we can combine these of course, but they measure different quantities.

Minor points:

1. Appendix table 6: you are reporting some very small numbers and proportions of participants (fewer than 5 in many cells). Is it ethical to release this information in a publication? Could participants be identified?

2. Figure 3, title: what is "health utilisation"? If the authors mean healthcare utilisation, this should be revised. Similarly, the title of appendix table: what is the "association between multimorbidity and outpatient"? Again, what are "inpatient number" and "outpatient number" in Appendix table 12 - numbers of patients or numbers of appointments? A previous reviewer (#2, comment 10) encouraged the authors to check the language throughout the manuscript, which the authors responded they had done. This appears to have been an overstatement.

3. Formatting of the appendix tables 8-11 needs to be revised for clarity.

4. Your responses to a number of comments from the previous reviewers (e.g. language, ways to examine the impact having vs. not having health insurance, and the grouping and /or labelling of multimorbidity categories - please see my major comment 2) were incompletely addressed.

Reviewer #3: Thanks for the revision, I have no further comments.

---

* Please upload any figures associated with your paper as individual TIF or EPS files with 300dpi resolution at resubmission; please read our figure guidelines for more information on our requirements: http://journals.plos.org/plosmedicine/s/figures. While revising your submission, we strongly recommend that you use PLOS's NAAS tool (https://ngplosjournals.pagemajik.ai/artanalysis) to test your figure files. NAAS can convert your figure files to the TIFF file type and meet basic requirements (such as print size, resolution), or provide you with a report on issues that do not meet our requirements and that NAAS cannot fix.

After uploading your figures to PLOS's NAAS tool - https://ngplosjournals.pagemajik.ai/artanalysis, NAAS will process the files provided and display the results in the "Uploaded Files" section of the page as the processing is complete.

If the uploaded figures meet our requirements (or NAAS is able to fix the files to meet our requirements), the figure will be marked as "fixed" above. If NAAS is unable to fix the files, a red "failed" label will appear above.

When NAAS has confirmed that the figure files meet our requirements, please download the file via the download option, and include these NAAS processed figure files when submitting your revised manuscript.

* Please ensure that the study is reported according to the appropriate guideline and include the completed checklist as Supporting Information. When completing the checklist, please use section and paragraph numbers, rather than page numbers. Please add the following statement, or similar, to the Methods: "This study is reported as per [XXXX] guideline (S1 Checklist)."

FIGURES AND TABLES

SUPPLEMENTARY MATERIAL

REFERENCES

STUDY TYPE-SPECIFIC REQUESTS:

OBSERVATIONAL STUDIES

* Abstract: Please include the study design, population and setting, number of participants, years during which the study took place (enrollment and follow up), length of follow up, and main outcome measures.

* Please ensure that the study is reported according to the STROBE (or appropriate STOBE extension) guideline (available from: https://www.equator-network.org/reporting-guidelines/strobe) and include the completed STROBE (or STROBE extension) checklist as Supporting Information. Please add the following statement, or similar, to the Methods: "This study is reported as per the Strengthening the Reporting of Observational Studies in Epidemiology (STROBE) guideline (S1 Checklist)." When completing the checklist, please use section and paragraph numbers, rather than page numbers.

* [FOR POPULATION HEALTH/REGISTRY STUDIES] Please ensure that the study is reported according to the RECORD guideline (available from https://www.record-statement.org) and include the completed checklist as Supporting Information. Please add the following statement, or similar, to the Methods: "This study is reported as per the Reporting of Studies Conducted using Observational Routinely-Collected Data (RECORD) guideline (S1 Checklist)." When completing the checklist, please use section and paragraph numbers, rather than page numbers.

* [FOR POPULATION HEALTH ESTIMATES] Please ensure that the study is reported according to the GATHER statement (available from https://www.equator-network.org/reporting-guidelines/gather-statement) and include the completed checklist as Supporting Information. Please add the following statement, or similar, to the Methods: "This study is reported as per the Guidelines for Accurate and Transparent Health Estimates Reporting (GATHER) statement (S1 Checklist)." When completing the checklist, please use section and paragraph numbers, rather than page numbers.

* [FOR MEDIATION ANALYSES] We recommend that the study is reported according to the AGReMA statement (https://agrema-statement.org/#:~:text=AGReMA%20is%20an%20evidence%2D%20and,randomised%20trials%20and%20observational%20studies) and include the completed checklist as Supporting Information. Please add the following statement, or similar, to the Methods: "This study is reported as per the Guideline for Reporting Mediation Analyses (AGReMA) statement (S1 Checklist)." When completing the checklist, please use section and paragraph numbers, rather than page numbers.

* For all observational studies, in the manuscript text, please indicate: (1) the specific hypotheses you intended to test, (2) the analytical methods by which you planned to test them, (3) the analyses you actually performed, and (4) when reported analyses differ from those that were planned, transparent explanations for differences that affect the reliability of the study's results. If a reported analysis was performed based on an interesting but unanticipated pattern in the data, please be clear that the analysis was data driven.

* Please state in the Methods section whether the study had a prospective protocol or analysis plan. If a prospective analysis plan (from your funding proposal, IRB or other ethics committee submission, study protocol, or other planning document written before analyzing the data) was used in designing the study, please include the relevant document(s) with your revised manuscript as a Supporting Information file to be published alongside your study and cite it in the Methods section. A legend for this file should be included at the end of your manuscript. If no such document exists, please make sure that the Methods section transparently describes when analyses were planned, and when/why any data-driven changes to analyses took place. Changes in the analysis, including those made in response to peer review comments, should be identified as such in the Methods section of the paper, with rationale.

MODELLING STUDIES

The following list is derived from Geoffrey P Garnett, Simon Cousens, Timothy B Hallett, Richard Steketee, Neff Walker. Mathematical models in the evaluation of health programmes. (2011) Lancet DOI:10.1016/S0140-6736(10)61505-X:

* If pertinent, please provide a diagram that shows the model structure, including how the natural history of the disease is represented, the process and determinants of disease acquisition, and how the putative intervention could affect the system.

* Please provide a complete list of model parameters, including clear and precise descriptions of the meaning of each parameter, together with the values or ranges for each, with justification or the primary source cited and important caveats about the use of these values noted.

* Please provide a clear statement about how the model was fitted to the data, including goodness-of-fit measure, the numerical algorithm used, which parameter varied, constraints imposed on parameter values, and starting conditions.

* For uncertainty analyses, please state the sources of uncertainties quantified and not quantified [can include parameter, data, and model structure].

* Please provide sensitivity analyses to identify which parameter values are most important in the model. Uncertainty estimates seek to derive a range of credible results on the basis of an exploration of the range of reasonable parameter values. The choice of method should be presented and justified.

* Please discuss the scientific rationale for the choice of model structure and identify points where this choice could influence conclusions drawn. Please also describe the strength of the scientific basis underlying the key model assumptions.

* For studies that develop a prediction model or evaluate its performance, please ensure that the study is reported according to the TRIPOD statement (https://www.equator-network.org/reporting-guidelines/tripod-statement) and include the completed checklist as Supporting Information. Please add the following statement, or similar, to the Methods: "This study is reported as per the Transparent Reporting of a Multivariable Prediction Model for Individual Prognosis Or Diagnosis (TRIPOD) statement (S1 Checklist)." For studies using machine learning, please use the TRIPOD-AI checklist. When completing the checklist, please use section and paragraph numbers, rather than page numbers.

SURVEY-BASED STUDIES

* Please ensure that the study is reported according to the CROSS guideline (https://www.equator-network.org/reporting-guidelines/a-consensus-based-checklist-for-reporting-of-survey-studies-cross/) and include the completed CROSS checklist as Supporting Information. Please add the following statement, or similar, to the Methods: "This study is reported as per A Consensus-Based Checklist for Reporting of Survey Studies (CROSS) guideline (S1 Checklist)." When completing the checklist, please use section and paragraph numbers, rather than page numbers.

* Please report your survey response rates according to AAPOR recommendations (https://aapor.org/standards-and-ethics/best-practices/)

* Please define how the population surveyed was sampled.

* Please compare characteristics of respondents and nonrespondents if possible.

* If sequential waves of the survey were sent, please specify whether the characteristics of respondents changed over time or remained constant.

* Please include the survey response rate in the Abstract.

* Please include a copy of the survey in the supplementary files.

HEALTH ECONOMICS / COST-EFFECTIVENESS STUDIES

* Please ensure that the study is reported according to the CHEERS guideline (available from: https://www.equator-network.org/reporting-guidelines/cheers) and include the completed checklist as Supporting Information. Please add the following statement, or similar, to the Methods: "This study is reported as per the Strengthening the Consolidated Health Economic Evaluation Reporting Standards 2022 (CHEERS 2022) Statement (S1 Checklist)." When completing the checklist, please use section and paragraph numbers, rather than page numbers.

---

## [Decision Letter · Decision Letter 2]

9 Mar 2026

Dear Dr He,

Many thanks for submitting your manuscript "Multimorbidity, health service use, and health insurance by socioeconomic groups in 31 countries: a multi-cohort study" (PMEDICINE-D-25-04109R2) to PLOS Medicine. The paper has been reviewed by subject experts and a statistician; their comments are included below and can also be accessed here: [LINK]

As you will see, the reviewer raised some comments about your additional analysis and requires further clarification. After discussing the paper with the editorial team and an academic editor with relevant expertise, I'm pleased to invite you to revise the paper in response to the reviewers' comments. We plan to send the revised paper to some or all of the original reviewers, and we cannot provide any guarantees at this stage regarding publication.

I have also added a list of editorial requests, so please check all points carefully (not all might apply) and make changes to your manuscript if apply.

We ask that you submit your revision by Mar 27 2026 11:59PM. However, if this deadline is not feasible, please contact me by email, and we can discuss a suitable alternative.

Don't hesitate to contact me directly with any questions (kstarowicz@plos.org). Between 14th and 28th of March I will be OOO travelling and attending conference in Japan. My response to emails will be delayed so in case of any urgent questions please contact plosmedicine@plos.org

Best regards,

Katarzyna

Katarzyna Starowicz, PhD

Associate Editor

PLOS Medicine

kstarowicz@plos.org

Comments from the reviewers:

Reviewer #1: I thank the authors for their careful and substantive responses to my previous comments. The revision has clearly strengthened the methodological transparency and analytical robustness of the manuscript. I appreciate the additional sensitivity analyses, clarifications of modelling assumptions, and improvements to reporting.

The following further comments are intended to enhance statistical clarity and interpretability.

Major Comments

1) The manuscript would benefit from a clearer articulation of the target estimand in the pooled meta-analytic analyses. While pooled estimates are now framed as descriptive summaries in the presence of substantial heterogeneity, it remains unclear whether the intended interpretation is a population-average effect across countries or merely a summary of country-specific associations.

2) Although multilevel mixed-effects models are presented, the manuscript does not report key variance components (e.g., random slope variance, intraclass correlation coefficients). Quantifying between-country variance would allow readers to assess the relative contribution of contextual versus individual-level factors and better interpret cross-national heterogeneity.

3) Model diagnostics are partially addressed (e.g., overdispersion, convergence), but additional evidence of model adequacy would improve transparency. Reporting comparative fit statistics (e.g., AIC/BIC across model families), marginal and conditional R² for mixed models, and brief summaries of residual diagnostics would enhance confidence in the chosen specifications.

4) The handling of missing data requires further clarification. The manuscript should explicitly report the extent and pattern of missingness across covariates and outcomes, the assumed missing data mechanism, and whether complete-case analysis or multiple imputation was employed. Given the cross-national and multi-cohort nature of the data, sensitivity analyses under alternative missingness assumptions would strengthen robustness.

5) The issue of multiplicity remains insufficiently addressed. The study includes multiple multimorbidity categories, several utilisation outcomes, country-specific analyses, interaction terms, and meta-analytic summaries. The authors should discuss the risk of type I error inflation and clarify whether any formal or interpretive adjustments were considered.

6) For count outcomes, while hurdle models were implemented as sensitivity analyses, the manuscript does not report the proportion of zero observations by country nor provide a clear justification for retaining the primary negative binomial specification over alternative frameworks. A more explicit comparison of model performance would strengthen methodological credibility.

7) The treatment of socioeconomic status (SES) warrants further consideration. If SES is categorised (e.g., wealth quartiles), the manuscript should clarify whether linear trends were tested or whether non-linearity was explored. The choice of functional form may materially influence effect estimates and equity interpretations.

8) The interpretation of healthcare utilisation as an indicator of economic burden should be treated more cautiously. Visit counts and admission numbers are imperfect proxies for economic impact, particularly in the absence of harmonised cost data across countries with different service intensity and financing structures.

Minor Comments

9) It would improve interpretability if incidence rate ratios were supplemented with marginal predicted counts (e.g., expected additional outpatient visits per year for individuals with multimorbidity). Translating relative measures into absolute differences would increase policy relevance.

10) Given cross-country variation in recall periods for healthcare utilisation, the manuscript should clarify whether utilisation measures were standardised (e.g., annualised) prior to modelling and how differences in survey wording may affect comparability.

11) The manuscript would benefit from a clearer distinction between formal insurance coverage and effective financial protection (e.g., cost-sharing, benefit depth), particularly when interpreting insurance-related effect modification in countries with near-universal coverage.

12) Cross-level interaction analyses between multimorbidity and macro-level system indicators (e.g., UHC index, health expenditure per capita) could be presented more formally within the multilevel framework, rather than relying solely on post hoc meta-regression.

---

* At this stage, we ask that you include a short, non-technical Author Summary of your research to make findings accessible to a wide audience that includes both scientists and non-scientists. The Author Summary should immediately follow the Abstract in your revised manuscript. This text is subject to editorial change and should be distinct from the scientific abstract. Ideally each sub-heading should contain 2-3 single sentence, concise bullet points containing the most salient points from your study. In the final bullet point of ‘What Do These Findings Mean?’ Please include the main limitations of the study in non-technical language.

Please see our author guidelines for more information: https://journals.plos.org/plosmedicine/s/revising-your-manuscript#loc-author-summary."

* Please confirm that your title complies with PLOS Medicine's style. Your title must be nondeclarative and not a question. It should begin with main concept if possible. "Effect of" should be used only if causality can be inferred, i.e., for an RCT. Please place the study design ("A randomized controlled trial," "A retrospective study," "A modelling study," etc.) in the subtitle (ie, after a colon).

* Please confirm that your abstract complies with our requirements, including format (three sections: Background, Methods and Findings, and Conclusions) and providing all the information relevant to this study type https://journals.plos.org/plosmedicine/s/submission-guidelines#loc-abstract

* Please ensure that the Introduction ends with a clear description of the study question or hypothesis.

* Please ensure that all abbreviations are defined at first use throughout the text.

* Please confirm that all numbers presented in the abstract are present and identical to numbers presented in the main manuscript text.

* Please review your text for claims of novelty or primacy (e.g. 'for the first time') and remove this language. In addition, please check that any use of statistical terms (such as trend or significant) are supported by the data, and if not please remove them.

* Please remove the 'conclusions' subheading from the discussion. Please also remove any other subheadings from the discussion.

* Statistical reporting: Please revise throughout the manuscript, including tables and figures.

- Please report statistical information as follows to improve clarity for the reader " "22% (95% CI [13,28]; p</=)" ".

- Please separate upper and lower bounds with commas instead of hyphens as the latter can be confused with reporting of negative values.

- Please repeat statistical definitions (HR, CI etc.) for each set of parentheses.

* In the abstract, please include the important dependent variables that are adjusted for in the analyses.

* In the author summary, in the final bullet point of 'What Do These Findings Mean?', please include the main limitations of the study in non-technical language.

* Please replace "subject" with participant, patient, individual, or person.

* The funding statement should include: specific grant numbers, initials of authors who received each award, URLs to sponsors’ websites. Also, please state whether any sponsors or funders (other than the named authors) played any role in study design, data collection and analysis, the decision to publish, or preparation of the manuscript. If they had no role in the research, include this sentence: “The funders had no role in study design, data collection and analysis, decision to publish, or preparation of the manuscript.”

* All authors must declare their relevant competing interests per the PLOS policy, which can be seen here: https://journals.plos.org/plosmedicine/s/competing-interests For authors with ties to industry, please indicate whether any of the interests has a financial stake in the results of the current study.

* PLOS defines the “minimal data set” to consist of the data set used to reach the conclusions drawn in the manuscript with related metadata and methods, and any additional data required to replicate the reported study findings in their entirety. Authors do not need to submit their entire data set, or the raw data collected during an investigation. Please submit the following data:

The values behind the means, standard deviations and other measures reported;

The values used to build graphs;

The points extracted from images for analysis.

* The Data Availability Statement (DAS) requires revision. For each data source used in your study:

"

* Thank you for agreeing to make your data available. At this time, please provide the link to the data repository and accession numbers required for access.

* Please provide the name(s) of the institutional review board(s) that provided ethical approval.

* Please specify whether informed consent was written or oral. Please ensure that the research complies with the PLOS policy in full: https://journals.plos.org/plosmedicine/s/human-subjects-research#loc-patient-privacy-and-informed-consent-for-publication

* Please define all elements of box plots in the figure caption - center line, box limits and whiskers.

* Please provide titles and legends for all figures and tables (including those in Supporting Information files). Please define all acronyms used in each figure or table in its corresponding legend.

* Please ensure that where relevant figures include 95% CIs.

* Please show graph axes beginning at zero. If this is not possible, please show a break in the axis.

* When a p value is given, please specify the statistical test used to determine it in the legend.

* Please convert any stacked bar charts to another data representation for example a table, or other type of graph.

* Please consider avoiding the use of red and green in order to make your figure more accessible

* Where data points are discrete, please ensure that they are depicted in the figures as discrete data and not as a continuous line.

* Please provide the unadjusted comparisons as well as the adjusted comparisons in all relevant Tables

* Please specify the variables controlled for in all relevant Tables

* Please include an acknowledgment of study participants.

* Please include in the Acknowledgments recognition of individuals who played a role in data collection or participant care or involvement

* Please upload any figures associated with your paper as individual TIF or EPS files with 300dpi resolution at resubmission; please read our figure guidelines for more information on our requirements: http://journals.plos.org/plosmedicine/s/figures. While revising your submission, we strongly recommend that you use PLOS's NAAS tool (https://ngplosjournals.pagemajik.ai/artanalysis) to test your figure files. NAAS can convert your figure files to the TIFF file type and meet basic requirements (such as print size, resolution), or provide you with a report on issues that do not meet our requirements and that NAAS cannot fix.

After uploading your figures to PLOS's NAAS tool - https://ngplosjournals.pagemajik.ai/artanalysis, NAAS will process the files provided and display the results in the "Uploaded Files" section of the page as the processing is complete.

If the uploaded figures meet our requirements (or NAAS is able to fix the files to meet our requirements), the figure will be marked as "fixed" above. If NAAS is unable to fix the files, a red "failed" label will appear above.

When NAAS has confirmed that the figure files meet our requirements, please download the file via the download option, and include these NAAS processed figure files when submitting your revised manuscript.

* Please ensure that the study is reported according to the [XXXX] guideline and include the completed [XXXX] checklist as Supporting Information. When completing the checklist, please use section and paragraph numbers, rather than page numbers. Please add the following statement, or similar, to the Methods: "This study is reported as per [XXXX] guideline (S1 Checklist)."

FIGURES AND TABLES

SUPPLEMENTARY MATERIAL

REFERENCES

STUDY TYPE-SPECIFIC REQUESTS:

OBSERVATIONAL STUDIES

* Abstract: Please include the study design, population and setting, number of participants, years during which the study took place (enrollment and follow up), length of follow up, and main outcome measures.

* Please ensure that the study is reported according to the STROBE (or appropriate STOBE extension) guideline (available from: https://www.equator-network.org/reporting-guidelines/strobe) and include the completed STROBE (or STROBE extension) checklist as Supporting Information. Please add the following statement, or similar, to the Methods: "This study is reported as per the Strengthening the Reporting of Observational Studies in Epidemiology (STROBE) guideline (S1 Checklist)." When completing the checklist, please use section and paragraph numbers, rather than page numbers.

* [FOR POPULATION HEALTH/REGISTRY STUDIES] Please ensure that the study is reported according to the RECORD guideline (available from https://www.record-statement.org) and include the completed checklist as Supporting Information. Please add the following statement, or similar, to the Methods: "This study is reported as per the Reporting of Studies Conducted using Observational Routinely-Collected Data (RECORD) guideline (S1 Checklist)." When completing the checklist, please use section and paragraph numbers, rather than page numbers.

* [FOR POPULATION HEALTH ESTIMATES] Please ensure that the study is reported according to the GATHER statement (available from https://www.equator-network.org/reporting-guidelines/gather-statement) and include the completed checklist as Supporting Information. Please add the following statement, or similar, to the Methods: "This study is reported as per the Guidelines for Accurate and Transparent Health Estimates Reporting (GATHER) statement (S1 Checklist)." When completing the checklist, please use section and paragraph numbers, rather than page numbers.

* [FOR MEDIATION ANALYSES] We recommend that the study is reported according to the AGReMA statement (https://agrema-statement.org/#:~:text=AGReMA%20is%20an%20evidence%2D%20and,randomised%20trials%20and%20observational%20studies) and include the completed checklist as Supporting Information. Please add the following statement, or similar, to the Methods: "This study is reported as per the Guideline for Reporting Mediation Analyses (AGReMA) statement (S1 Checklist)." When completing the checklist, please use section and paragraph numbers, rather than page numbers.

* For all observational studies, in the manuscript text, please indicate: (1) the specific hypotheses you intended to test, (2) the analytical methods by which you planned to test them, (3) the analyses you actually performed, and (4) when reported analyses differ from those that were planned, transparent explanations for differences that affect the reliability of the study's results. If a reported analysis was performed based on an interesting but unanticipated pattern in the data, please be clear that the analysis was data driven.

* Please state in the Methods section whether the study had a prospective protocol or analysis plan. If a prospective analysis plan (from your funding proposal, IRB or other ethics committee submission, study protocol, or other planning document written before analyzing the data) was used in designing the study, please include the relevant document(s) with your revised manuscript as a Supporting Information file to be published alongside your study and cite it in the Methods section. A legend for this file should be included at the end of your manuscript. If no such document exists, please make sure that the Methods section transparently describes when analyses were planned, and when/why any data-driven changes to analyses took place. Changes in the analysis, including those made in response to peer review comments, should be identified as such in the Methods section of the paper, with rationale.

* Your study is observational and therefore causality cannot be inferred. Please remove language that implies causality and refer to associations instead.

* For all observational studies, in the manuscript text, please indicate: (1) the specific hypotheses you intended to test, (2) the analytical methods by which you planned to test them, (3) the analyses you actually performed, and (4) when reported analyses differ from those that were planned, transparent explanations for differences that affect the reliability of the study's results. If a reported analysis was performed based on an interesting but unanticipated pattern in the data, please be clear that the analysis was data-driven.

MODELLING STUDIES

The following list is derived from Geoffrey P Garnett, Simon Cousens, Timothy B Hallett, Richard Steketee, Neff Walker. Mathematical models in the evaluation of health programmes. (2011) Lancet DOI:10.1016/S0140-6736(10)61505-X:

* If pertinent, please provide a diagram that shows the model structure, including how the natural history of the disease is represented, the process and determinants of disease acquisition, and how the putative intervention could affect the system.

* Please provide a complete list of model parameters, including clear and precise descriptions of the meaning of each parameter, together with the values or ranges for each, with justification or the primary source cited and important caveats about the use of these values noted.

* Please provide a clear statement about how the model was fitted to the data, including goodness-of-fit measure, the numerical algorithm used, which parameter varied, constraints imposed on parameter values, and starting conditions.

* For uncertainty analyses, please state the sources of uncertainties quantified and not quantified [can include parameter, data, and model structure].

* Please provide sensitivity analyses to identify which parameter values are most important in the model. Uncertainty estimates seek to derive a range of credible results on the basis of an exploration of the range of reasonable parameter values. The choice of method should be presented and justified.

* Please discuss the scientific rationale for the choice of model structure and identify points where this choice could influence conclusions drawn. Please also describe the strength of the scientific basis underlying the key model assumptions.

* For studies that develop a prediction model or evaluate its performance, please ensure that the study is reported according to the TRIPOD statement (https://www.equator-network.org/reporting-guidelines/tripod-statement) and include the completed checklist as Supporting Information. Please add the following statement, or similar, to the Methods: "This study is reported as per the Transparent Reporting of a Multivariable Prediction Model for Individual Prognosis Or Diagnosis (TRIPOD) statement (S1 Checklist)." For studies using machine learning, please use the TRIPOD-AI checklist. When completing the checklist, please use section and paragraph numbers, rather than page numbers.

HEALTH ECONOMICS / COST-EFFECTIVENESS STUDIES

* Please ensure that the study is reported according to the CHEERS guideline (available from: https://www.equator-network.org/reporting-guidelines/cheers) and include the completed checklist as Supporting Information. Please add the following statement, or similar, to the Methods: "This study is reported as per the Strengthening the Consolidated Health Economic Evaluation Reporting Standards 2022 (CHEERS 2022) Statement (S1 Checklist)." When completing the checklist, please use section and paragraph numbers, rather than page numbers.

---

## [Decision Letter · Decision Letter 3]

20 Apr 2026

Dear Dr. He,

Thank you very much for re-submitting your manuscript "Multimorbidity, health service use, and health insurance by socioeconomic groups in 31 countries: a multi-cohort study" (PMEDICINE-D-25-04109R3) for review by PLOS Medicine.

I have discussed the paper with my colleagues and the academic editor and it was also seen again by reviewers. I am pleased to say that provided the remaining editorial and production issues are dealt with we are planning to accept the paper for publication in the journal.

The remaining issues that need to be addressed are listed at the end of this email. Please ensure that you check all of them carefully and provide response. Any accompanying reviewer attachments can be seen via the link below. Please take these into account before resubmitting your manuscript:

[LINK]

We look forward to receiving the revised manuscript by Apr 27 2026 11:59PM.

Sincerely,

Katarzyna Starowicz, PhD

Associate Editor

PLOS Medicine

kstarowicz@plos.org

Requests from Editors:

* Please, ensure that you uploaded clean version of manuscript (no tracked version like the one sent by email to me) and incorporate response to editor comments file in it (you can combine it with response to reviewers file; please include responses sent by email to previous decisions and from this one)

* Statistical reporting: Please revise throughout the manuscript, including tables and figures.

- Please report statistical information as follows to improve clarity for the reader " "22% (95% CI [13,28]; p</=)" ".

- Please separate upper and lower bounds with commas instead of hyphens as the latter can be confused with reporting of negative values.

- Please repeat statistical definitions (HR, CI etc.) for each set of parentheses.

* The Github repository link is working but it seems that repository is empty, please fix it

* Please remove the subheadings from the discussion.

* Please review your text for claims of novelty or primacy (e.g. 'for the first time') and remove this language. In addition, please check that any use of statistical terms (such as trend or significant) are supported by the data, and if not please remove them.

* Please ensure that the study is reported according to the STROBE guideline and include the completed STROBE checklist as Supporting Information. When completing the checklist, please use section and paragraph numbers, rather than page numbers. Please add the following statement, or similar, to the Methods: "This study is reported as per STROBE guideline (S1 Checklist)."

* Please ensure that the Introduction ends with a clear description of the study question or hypothesis.

Comments from Reviewers:

Reviewer #1: The authors have satisfactorily addressed my previous concerns. The additional analyses (multilevel models, sensitivity analyses, and interaction testing) strengthen the robustness of the findings. Remaining limitations are appropriately acknowledged. I support publication.

[LINK]

---

## [Editor Report · Decision Letter 4]

23 Apr 2026

Dear Dr He,

On behalf of my colleagues and the Academic Editor, Margaret E Kruk, I am pleased to inform you that we have agreed to publish your manuscript "Multimorbidity, health service use, and health insurance by socioeconomic groups in 31 countries: a multi-cohort study" (PMEDICINE-D-25-04109R4) in PLOS Medicine.

PRESS

Sincerely,

Katarzyna Starowicz, PhD

Associate Editor

PLOS Medicine